# A Computationally Viable Numerical Gradient-based Technique for Optimal Covering Problems

**Gokul Rajaraman**
Department of Mechanical Engineering
IIT Bombay
Mumbai 400076, India
22b4517@iitb.ac.in

**Debasish Chatterjee**
Centre for Systems and Control
IIT Bombay
Mumbai 400076, India
dchatter@iitb.ac.in

## Abstract

The problem of optimally covering a given compact subset of $\mathbb{R}^N$ with a preassigned number $n$ of Euclidean metric balls has a long-standing history and it is well-recognized to be computationally hard. This article establishes a numerically viable algorithm for obtaining optimal covers of compact sets via two key contributions. The first is a foundational result establishing Lipschitz continuity of the marginal function of a certain parametric non-convex maximization problem in the optimal covering problem, and it provides the substrate for numerical gradient algorithms to be employed in this context. The second is an adaptation of a stochastically smoothed numerical gradient-based (zeroth-order) algorithm for a non-convex minimization problem, that, equipped with randomized restarts, spurs global convergence to an optimal cover. Several numerical experiments with complicated nonconvex compact sets demonstrate the excellent performance of our techniques.

## 1 Introduction

Optimal covering of sets is an important problem that arises naturally in several fields of science and technology, including learning and approximation theory, computer science, signal processing, information theory, combinatorics, communication, sensor networks, multi-agent systems, cyberphysical systems, etc. In a sensor network, for instance, given a collection of sensors having preassigned sensing radius and a set that must be enveloped by the sensors, an important engineering question is to find an optimal collection of points for placing the sensors to ensure complete coverage of the given set. An identical problem arises for the coverage of a given geographical region with cell-phone towers in order to maximize network availability in the region. In information theory, assuming that one can accurately decode messages from a certain set within a preassigned maximal error tolerance for each, it is important to determine the number of 'signals' that are necessary for accurate decoding of all the messages; this is once again a problem of optimal covering of the set of messages with a collection of 'signals'.

The covering problem appears in various forms, and perhaps its most common variant is captured by the following situation. For a given non-empty (and possibly uncountable) compact set $\mathcal{M} \subset \mathbb{R}^N$ and a preassigned $\varepsilon > 0$, one finds a finite set $\mathcal{F} \subset \mathbb{R}^N$ that is at most $\varepsilon$-distant from any point in $\mathcal{M}$. Such a minimal set is an $\varepsilon$-*net* and its cardinality is the so-called $\varepsilon$-*covering number* of the set $\mathcal{M}$; see, e.g., (Kolmogorov and Tihomirov, 1961) for an early detailed exposé of related topics and (Alimov and Tsar'kov, 2021, Chapters 15 and 16) for a more recent treatment. A closely related and equally relevant variant of covering is its resource-constrained or budgeted version: Given a number $n$ of metric balls, one must find the minimal radius and a placement of the $n$ centers of the balls such that their union covers $\mathcal{M}$. Such a collection of the centers is commonly known as a *Chebyshev net*

*of cardinality $n$* (Alimov and Tsar'kov, 2021, Definition 15.3).[1] We address this particular problem in our current work.

Formally, given a non-empty and compact $\mathcal{M} \subset \mathbb{R}^N$, we seek to solve the optimization problem

$$\epsilon(\mathcal{M}, n) := \min_{(x_1,\ldots,x_n)\in\mathbb{R}^{N\times n}} \max_{y\in\mathcal{M}} \min_{i=1,\ldots,n} \|y - x_i\|. \tag{1}$$

For an $n$-tuple of centers $(x_1,\ldots,x_n) \in \mathbb{R}^{N\times n}$ and $y \in \mathcal{M}$, $\min_{i=1,\ldots,n}\|y - x_i\|$ determines the distance of $y$ from $\bigcup_{i=1}^n \{x_i\}$. Consequently, the number $\max_{y\in\mathcal{M}} \min_{i=1,\ldots,n}\|y - x_i\|$ is the maximum distance of points in $\mathcal{M}$ from the collection $\bigcup_{i=1}^n \{x_i\}$, and the outer minimization in (1) optimizes this maximum distance over the centers $(x_1,\ldots,x_n)$. The quantity $\epsilon(\mathcal{M}, n)$ is known as the *covering radius* of the set $\mathcal{M}$ with $n$ metric balls as defined in (Alimov and Tsar'kov, 2021, p. 300); solving (1) yields an optimal $n$-point representation — a *Chebyshev $n$-net* — of $\mathcal{M}$, and $\epsilon(\mathcal{M}, n)$ is the error guarantee in such a representation.

Chebyshev $n$-nets are especially useful for estimating errors in function learning in the presence of certain regularity properties. For instance, consider a target function $f : \mathcal{M} \to \mathbb{R}$ that is known to be Lipschitz continuous with modulus $L_0$ but is otherwise unknown, and suppose that a Chebyshev $n$-net $\mathcal{F}$ of $\mathcal{M}$ and the set of values $\{f(x) \mid x \in \mathcal{F}\}$ of $f$ on $\mathcal{F}$ are available. Then a tight estimate of the value of $f$ at a generic $y \in \mathcal{M}$ may be obtained, within an absolute error margin of $L_0\epsilon(\mathcal{M}, n)$, by finding $y_\star \in \arg\min_{z\in\mathcal{F}} \|z - y\|$ and then calculating $f(y_\star)$. By an identical reasoning, for Hölder continuous $f$ of modulus $C > 0$ and rate $\alpha \in {]}0, 1{[}$, the corresponding error bound is $C\epsilon(\mathcal{M}, n)^\alpha$. Similar arguments in optimization problems involving uniformly continuous objective functions permit the derivation of quantitative estimates of true optima from estimates over finitely many points.

In several applications of covering, the centers of the metric balls may be restricted to lie within $\mathcal{M}$, in which case, the problem (1) should be modified to

$$\epsilon'(\mathcal{M}, n) := \min_{(x_1,\ldots,x_n)\in\mathcal{M}^n} \max_{y\in\mathcal{M}} \min_{i=1,\ldots,n} \|y - x_i\|. \tag{2}$$

This constrained version is relevant in settings where the representative points must strictly belong to the set under consideration. The triangle inequality immediately yields the sandwich relation $\frac{\epsilon'(\mathcal{M},n)}{2} \leqslant \epsilon(\mathcal{M}, n) \leqslant \epsilon'(\mathcal{M}, n)$. Our efforts are focused on the numerical viability of (1) for compact $\mathcal{M}$; similar methods can be adapted with minor modifications to address (2) if $\mathcal{M}$ is convex.

**Contributions**

The problem (1) is challenging (NP-hard) even in the case $n = 1$. Indeed, this situation is the so-called Chebyshev center problem (Alimov and Tsar'kov, 2021, p. 296) and is of central importance in optimal function learning (Binev et al., 2024), signal processing (Micchelli and Rivlin, 1977; Eldar et al., 2008; Wu et al., 2013), approximation theory (Alimov and Tsar'kov, 2019; Foucart and Liao, 2024), and a host of other disciplines. Despite its complexity, it turns out that the Chebyshev center problem admits a numerically viable solution technique (with excellent approximation properties) by leveraging certain convexity attributes present therein, and has been recently studied in (Paruchuri and Chatterjee, 2023). The case of $n > 1$ considered in the current article is significantly more difficult: of course, it is challenging (NP-hard), and in addition, it does not appear to be amenable to any simplification via convexity.[2] Moreover, while the Chebyshev center of $\mathcal{M}$ is unique for the Euclidean norm, the Chebyshev $n$-net problem may not yield unique solutions.[3]

Since (1) is well-known to be computationally challenging, we focus attention on the development of viable algorithm based on ***numerical gradients*** (i.e., a ***zeroth-order*** algorithm) with good approximation properties to solve (1):

○ To this end, we demonstrate that the inner max-min problem in (1) admits a reformulation that yields good regularity properties parametrically in $(x_1,\ldots,x_n)$ corresponding to the outer minimization

---

[1]In sensor and communication networks, for instance, the radius of sensing/communication of each sensor is dependent on the power at hand, which lends a measure of flexibility in altering the indicated radius.

[2]For instance, the Chebyshev center of $\mathcal{M}$ is identical to the Chebyshev center of the convex hull of $\mathcal{M}$, but that is false in general for Chebyshev $n$-nets for $n > 1$.

[3]In a few cases of $\mathcal{M}$, closed-form expressions for Chebyshev $n$-nets are known; see, e.g., (Ushakov and Lebedev, 2015).

under mild hypotheses; this is our first main result Theorem 3.2. The indicated regularity property enables and justifies the deployment of a numerical gradient-based (zeroth-order) algorithm in the variables $(x_1, \ldots, x_n)$ to arrive at a solution to (1) itself. To our knowledge, ours is the only instance of a gradient-based approach to finding Chebyshev $n$-nets for compact subsets of $\mathbb{R}^N$ that do not necessarily possess special structure.

○ In order to obtain the indicated regularity properties, the key technical tools employed here are derived from the field of sensitivity of optimization problems (Fiacco, 1983; Fiacco and Ishizuka, 1990) or variational analysis (Dontchev and Rockafellar, 2014). We establish the crucial ***Lipschitz continuity*** of the (parametric) inner max-min problem in (1) with respect to $(x_1, \ldots, x_n)$ around optimal points of the outer minimization; this lays down the foundation for the deployment of a range of numerical gradient-based algorithms for the outer minimization.

○ Numerical gradient-based algorithms for nonlinear programs are not expected to converge to global optimizers in general. Here we lift off-the-shelf a recent zeroth-order algorithm that ***stochastically smoothes*** the numerical gradients and ***explores*** the decision landscape in search for global optima; the precise complexity bound appears in our second main result Theorem 3.6. More refined algorithms involving, e.g., momenta for convergence to global optimizers are topics of subsequent investigations. This is the first instance, to the best of our knowledge, that powerful tools from sensitivity theory have been employed in conjunction with numerical gradient-based algorithms to address the Chebyshev $n$-net problem.

○ The efficacy of our algorithm depends on the availability of accurate solutions to the inner max-min problem (after an appropriate reformulation) for each fixed $(x_1, \ldots, x_n)$. The indicated parametric max-min problem happens to be non-convex despite our reformulation, and the speed of execution of our algorithm depends on how quickly these globally optimal solutions are made available. Our numerical experiments indicate that despite the lack of convexity, our algorithm (including resets) gives ***excellent performance*** even for relatively complicated ***nonconvex sets*** $\mathcal{M}$ — see §4 for details. Scalability of our algorithm with increasing dimensions $N$ and number $n$ of centers remains a challenging issue: while the numerical gradient-based algorithm for the outer minimization scales in these variables as $\mathcal{O}(N^{\frac{3}{2}} n^2)$, the inner max-min is non-convex and remains the chief computational bottleneck in our algorithm.

Since the Chebyshev $n$-net problem is understood to be hard, several randomized algorithms to *approximately* solve (1) have been developed; see, e.g., (Har-Peled, 2011, §5.3) for an account. While many of these randomized techniques involve independent and identically distributed sampling,[4] more complicated history-dependent stochastic iterative algorithms have been advanced in the literature for special cases — see, e.g., (Ushakov and Lebedev, 2015) (and the references therein) for the case of $N = 3$. Other techniques to solve the optimal $n$-net and related problems relevant to theoretical computer science and computational geometry may be found in (Malysheva, 2020; Yu, 2021; Voronov et al., 2023). It is, of course, possible to employ zeroth-order methods (such as simulated annealing (Robert and Casella, 2004)) that do not rely on any regularity of the inner max-min in (1). However, the Lipschitz regularity of the parametric max-min (Theorem 3.2) paves the way for fine-tuned algorithms that exploit this regularity compared to typically slow annealing techniques; as such, Theorem 3.2 constitutes the key point of departure of our contributions from the current literature.

**Organization**

This article unfolds as follows. Several preparatory results are contained in §2; they are employed in the two main results centered around our key Algorithm 1 in §3. Numerical experiments are presented in §4 and the technical data corresponding to both our theoretical results and numerical experiments are collected in §A–§D.

## 2 Preliminaries

It is intuitive to expect that an optimizer $(x_1^*, \ldots, x_n^*)$ of (1) should be restricted to a bounded set that contains $\mathcal{M}$. Recall that $\epsilon(\mathcal{M}, 1)$ is the *Chebyshev radius* of $\mathcal{M}$, and an optimizer $z^*$ of (1)

---

[4]Methods relying on independent and identical sampling typically suffer from sharply worsening performance as the dimension rises due to concentration of measure effects; see (Das et al., 2022) for details.

for $n = 1$ is the *Chebyshev center* of $\mathcal{M}$.[5] In other words, the Chebyshev center of $\mathcal{M}$ provides the best single-point representation of the set $\mathcal{M}$, minimizing the worst-case distance to all points in $\mathcal{M}$. There is a neat relationship between the $n$-net problem (1) and the Chebyshev center of $\mathcal{M}$ established in the following proposition, a proof of which is in §B.

**Proposition 2.1.** *If $(x_1^*, \ldots, x_n^*)$ is an optimizer of (1) and if $z^*$ is a Chebyshev center of $\mathcal{M}$, then*

$$\|z^* - x_i^*\| \leqslant 2\epsilon(\mathcal{M}, 1) \quad \text{for all } i \in \{1, \ldots, n\}. \tag{3}$$

*In other words, without loss of generality, all optimizers of (1) can be restricted to a ball of twice the Chebyshev radius, centered at the Chebyshev center. Henceforth, we denote this ball by $\mathbb{B}(\mathcal{M})$ whereby, it follows that $(x_1^*, \ldots, x_n^*) \in \mathbb{B}(\mathcal{M})^n$*

This result is useful for generating initial guesses and bounding our search space when implementing our algorithm, which will be discussed in subsequent sections. Inspection of (1) reveals that $\min_{i=1,\ldots,n} \|y - x_i\|$ is non-convex in $(x_1, \ldots, x_n)$ for fixed $y$ and vice versa.

As indicated in §1, we reformulate the problem in a way suitable for the construction of a numerically viable algorithm to extract near optimal solutions. To this end, we first note that the geometric implication of the problem guarantees strict feasibility for any compact $\mathcal{M}$. Moreover, for a fixed set of centers, the inner maximization always has a finite optimal value owing to the compactness of $\mathcal{M}$. We now present a straightforward reformulation of (1) using a slack variable, which, despite its simplicity, provides the problem with a richer structure for computational viability. Since the square of the norm ensures continuous differentiability, a property that will become important subsequently, we work with the squared norm by noting that

$$\min_{i=1,\ldots,n} \|y - x_i\|^2 = \max\{t \in \mathbb{R} \mid t \leqslant \|y - x_i\|^2 \text{ for all } i = 1, \ldots, n\},$$

which immediately leads to

$$
\begin{aligned}
\epsilon(\mathcal{M}, n)^2 = \min_{(x_1,\ldots,x_n) \in \mathbb{R}^{N \times n}} \max_{y \in \mathcal{M}} \max_{t \in \mathbb{R}} \quad & t \\
\text{subject to} \quad & t - \|y - x_i\|^2 \leqslant 0 \quad \text{for all } i = 1, \ldots, n.
\end{aligned}
\tag{4}
$$

It is natural to seek a unification of both inner maximizations into a single maximization over the Cartesian product of their respective domains. The following proposition, whose proof is in §B, establishes that this unification is valid for the given problem.

**Proposition 2.2.** *For a given set of centers $(x_1, \ldots, x_n) \in \mathbb{R}^{N \times n}$, if $(t^\dagger, y^\dagger)$ is an optimizer of*

$$\max\left\{\max\{t \in \mathbb{R} \mid t - \|y - x_i\|^2 \leqslant 0 \text{ for all } i = 1, \ldots, n\} \mid y \in \mathcal{M}\right\} \tag{5}$$

*and $(t^*, y^*)$ is an optimizer of*

$$\max\left\{t \mid (t, y) \in \mathbb{R} \times \mathcal{M} \text{ and } t - \|y - x_i\|^2 \leqslant 0 \text{ for all } i = 1, \ldots, n\right\}, \tag{6}$$

*then $t^\dagger = t^*$. That is, both (5) and (6) are feasible and have the same optimal value.*

In view of of Proposition 2.2, from this point forward, we adopt the following reformulation for the $n$-covering radius:

$$\epsilon(\mathcal{M}, n)^2 = \min_{(x_1,\ldots,x_n) \in \mathbb{R}^{N \times n}} \mathcal{G}(x_1, \ldots, x_n), \tag{7}$$

where

$$\mathcal{G}(x_1, \ldots, x_n) := \max\left\{t \mid (t, y) \in \mathbb{R} \times \mathcal{M} \text{ and } t - \|y - x_i\|^2 \leqslant 0 \text{ for all } i = 1, \ldots, n\right\}.$$

Evaluating $\mathcal{G}$ entails solving a nonconvex problem, and the gradient of $\mathcal{G}$ is difficult to calculate analytically, especially without imposing further structure on the inner maximization. There are several 'gradient-free' algorithms such as simulated annealing via Markov chain Monte Carlo to ensure probabilistic convergence to global minima, simplicial algorithms (e.g., the Nelder-Meade) for deterministic convergence to global minima, of $\mathcal{G}$, etc. Although these algorithms do not require further structure on $\mathcal{M}$ and may be easy to implement since they only employ point evaluations of

---

[5]The Chebyshev center is necessarily unique for us due to strict convexity of the Euclidean norm.

$\mathcal{G}$, numerical gradient/subgradient-based algorithms tend to furnish solutions more efficiently and scale well relative to the size of the problem data, provided the objective function satisfies certain regularity conditions. An algorithm centered around numerical gradients is our target.

To advance in this direction, we impose the condition that $\mathcal{M}$ is the intersection of 0-sublevel sets of finitely many known continuously differentiable functions in the following way. Let $\mathcal{O}_1, \ldots, \mathcal{O}_k \in \mathbb{R}^N$ be open sets containing $\mathcal{M}$, let $\varphi_j : \mathcal{O}_j \to \mathbb{R}$ for $j = 1, \ldots k$, be continuously differentiable functions, and suppose that $\mathcal{M}$ is realized as

$$\mathcal{M} = \left\{ y \in \mathbb{R}^N \mid \varphi_j(y) \leqslant 0 \text{ for } j = 1, \ldots, k \right\}.$$

This allows us to write $\mathcal{G}$ as the negative of the optimal value function of a standard parametric optimization problem

$$-\mathcal{G}(x_1, \ldots, x_n) = \min_{(t,y)} \quad -t$$

$$\text{subject to} \quad \begin{cases} t - \|y - x_i\|^2 \leqslant 0 & \text{for all } i = 1, \ldots, n, \\ \varphi_{i-n}(y) \leqslant 0 & \text{for all } i = n+1, \ldots, n+k, \\ t \geqslant 0, \\ (t, y) \in \mathbb{R} \times \mathbb{R}^N. \end{cases} \tag{8}$$

**Example 2.3.** The problem (8), for a given choice of $(x_1, \ldots, x_n)$, need not have strict local optimizers in general. To see this, let $\mathcal{M} \subset \mathbb{R}^2$ be a disc of radius $R > 0$ centered at the origin and let $n = 1$. The evaluation of $\mathcal{G}$ at the origin leads to non-isolated optimizers for (8) since any pair of the form $(R^2, y^*)$ with $\|y^*\| = R$ is an optimizer.

*Remark* 2.4. Note that it is not necessary to explicitly impose the condition $t \geqslant 0$ in (8), as the optimal value remains unchanged and is always positive. However, we shall see below that including this constraint ensures compactness of the feasible region — a property whose significance will be addressed subsequently. One can observe that if the functions $\varphi_j$ for $j = 1, \ldots, k$, are convex (which in turn implies that $\mathcal{M}$ is convex), then the problem (8) becomes a DC (Difference of Convex functions) program since the objective and the constraints are differences of two convex functions. The reader is referred to (An et al., 2014) for a comprehensive overview of DC programming and algorithms to solve problems with DC objectives and constraints upto criticality; see, e.g., (Mordukhovich, 2018, Chapter 7) for theoretical insights. Although our results do not require the functions to be convex, when they are, DC programming algorithms may be used to achieve faster convergence to criticality in numerical implementations.

## 3 Main results

We now outline some technical notations and definitions in the context of the problem (8) to set the stage for our analysis. The Lagrangian $\mathcal{L} : \mathbb{R}^{N \times n} \times (\mathbb{R} \times \mathbb{R}^N) \times [0, +\infty[^{n+k+1} \to \mathbb{R}$ of (8) is

$$\mathcal{L}(x, (t, y), \lambda) := -(\lambda_{n+k+1} + 1)t + \sum_{i=1}^{n} \lambda_i \left(t - \|y - x_i\|^2\right) + \sum_{i=n+1}^{n+k} \lambda_i \varphi_{i-n}(y), \tag{9}$$

where $x$ denotes $(x_1, \ldots, x_n)$.

The feasible region is denoted by the set-valued map $S_{\text{feas}} : \mathbb{R}^{N \times n} \rightrightarrows \mathbb{R} \times \mathbb{R}^N$,

$$S_{\text{feas}}(\bar{x}) := \left\{ (t, y) \in \mathbb{R} \times \mathbb{R}^N \;\middle|\; \begin{array}{ll} t - \|y - \bar{x}_i\|^2 \leqslant 0 & \text{for all } i = 1, \ldots, n, \\ \varphi_{i-n}(y) \leqslant 0 & \text{for all } i = n+1, \ldots, n+k, \\ t \geqslant 0 \end{array} \right\}, \tag{10}$$

and the set of global optimizers is the set-valued map $S_{\text{opt}} : \mathbb{R}^{N \times n} \rightrightarrows \mathbb{R} \times \mathbb{R}^N$,

$$S_{\text{opt}}(\bar{x}) := \left\{ (t, y) \in S_{\text{feas}}(\bar{x}) \mid t = -\mathcal{G}(\bar{x}) \right\}. \tag{11}$$

For a choice $\bar{x}$ of $x$, let $(\bar{t}, \bar{y}) \in S_{\text{opt}}(\bar{x})$. We define the set of active constraint indices at $(\bar{x}, (\bar{t}, \bar{y}))$ as

$$I(\bar{x}, (\bar{t}, \bar{y})) := \left\{ i \in \{1, \ldots, n\} \mid \bar{t} - \|\bar{y} - \bar{x}_i\|^2 = 0 \right\} \cup \left\{ i \in \{n+1, \ldots, n+k\} \mid \varphi_{i-n}(\bar{y}) = 0 \right\}. \tag{12}$$

The constraint $t \geqslant 0$ is not included since $t$ remains strictly positive at optimality. The set of Karush-Kuhn-Tucker (KKT) vectors of (8) for a pair $(\bar{x}, (\bar{t}, \bar{y})) \in \mathbb{R}^{N \times n} \times (\mathbb{R} \times \mathbb{R}^N)$ is denoted by

$$\Lambda(\bar{x}, (\bar{t}, \bar{y})) := \left\{ \lambda \in [0, +\infty[^{n+k+1} \left| \begin{array}{l} \nabla_{(t,y)} \mathcal{L}(\bar{x}, (\bar{t}, \bar{y}), \lambda) = 0, \\ \lambda_i \left( \bar{t} - \|\bar{y} - \bar{x}_i\|^2 \right) = 0 \text{ for } i = 1, \ldots, n \\ \lambda_i \varphi_{n-i}(\bar{y}) = 0 \text{ for } i = n+1, \ldots, n+k, \text{ and} \\ \lambda_{n+k+1} \bar{t} = 0 \end{array} \right. \right\} \quad (13)$$

**Definition 3.1** (MFCQ for (8)). For $x \in \mathbb{R}^{N \times n}$, the *Mangasarian-Fromovitz constraint qualification* condition is said to hold at $(\bar{t}, \bar{y}) \in S_{\text{opt}}(\bar{x})$ if there exists a direction $w \in \mathbb{R}^{N+1}$ such that the inner product of $w$ with the gradient of each active constraint is strictly negative. To wit, the following conditions must hold:

(3.1-i) $\begin{pmatrix} 1 & -2(\bar{y} - \bar{x}_i)^\top \end{pmatrix} w < 0$ for all $i \in \{1, \ldots, n\} \cap I(\bar{x}, (\bar{t}, \bar{y}))$, and

(3.1-ii) $\nabla_y \varphi_i(\bar{y})^\top w < 0$ for all $i \in \{n+1, \ldots, n+k\} \cap I(\bar{x}, (\bar{t}, \bar{y}))$.

We now present our first main result establishing Lipschitz continuity of $\mathcal{G}$.

**Theorem 3.2.** *Consider the problem (8) and grant the notations established above. Fix $\bar{x} \in \mathbb{R}^{N \times n}$ and suppose that the MFCQ condition (defined in Definition 3.1) holds for every $(\bar{t}, \bar{y}) \in S_{\text{opt}}(\bar{x})$. Then the function $\mathcal{G}$ is locally Lipschitz continuous at $\bar{x}$. Consequently, its restriction $\mathcal{G}|_{\mathbb{B}(\mathcal{M})^n}$ to $\mathbb{B}(\mathcal{M})^n$ (c.f. Proposition 2.1) is Lipschitz. Moreover, for any $v \in \mathbb{R}^{N \times n}$, we have*

$$\inf_{(\bar{t}, \bar{y}) \in S_{\text{opt}}(\bar{x})} \min_{\lambda \in \Lambda(\bar{x}, (\bar{t}, \bar{y}))} \frac{\partial \mathcal{L}}{\partial x}(\bar{x}, (\bar{t}, \bar{y}), \lambda) \cdot v \leqslant - \liminf_{h \downarrow 0} \frac{1}{h} (\mathcal{G}(\bar{x} + hv) - \mathcal{G}(\bar{x}))$$

$$\leqslant - \limsup_{h \downarrow 0} \frac{1}{h} (\mathcal{G}(\bar{x} + hv) - \mathcal{G}(\bar{x})) \leqslant \inf_{(\bar{t}, \bar{y}) \in S_{\text{opt}}(\bar{x})} \max_{\lambda \in \Lambda(\bar{x}, (\bar{t}, \bar{y}))} \frac{\partial \mathcal{L}}{\partial x}(\bar{x}, (\bar{t}, \bar{y}), \lambda) \cdot v. \quad (14)$$

*Proof.* We observe that both the objective and constraint functions are continuously differentiable. At each $x \in \mathbb{R}^{N \times n}$, the feasible region $S_{\text{feas}}(\bar{x})$ is non-empty by definition. Letting $N(\bar{x})$ denote a neighborhood of $\bar{x}$, we note that the mapping $S_{\text{feas}}$ is uniformly compact near $\bar{x}$ since the set $\bigcup_{x \in N(\bar{x})} S_{\text{feas}}(x)$ always satisfies the containment relation

$$\bigcup_{x \in N(\bar{x})} S_{\text{feas}}(x) \subset \left[ 0, \max_{x \in N(\bar{x})} \max_{y \in \mathcal{M}} \max_{i=1,\ldots,n} \|y - x_i\|^2 \right] \times \mathcal{M}.$$

Furthermore, a set independent of the base point $\bar{x}$ containing $\bigcup_{x \in N(\bar{x})} S_{\text{feas}}(x)$ can be realized if each component of $\bar{x}$ is restricted to the interior of the ball described in Proposition 2.1, in which case we obtain the uniform containment $\bigcup_{x \in N(\bar{x})} S_{\text{feas}}(x) \subset \left[ 0, 9\epsilon(\mathcal{M}, 1)^2 \right] \times \mathcal{M}$; the set on the right-hand side is, of course, bounded. Therefore, (Fiacco and Ishizuka, 1990, Theorem 4.2) — reproduced for completeness in §C as Theorem C.1 — applies directly to our problem since the MFCQ conditions hold for every $(\bar{t}, \bar{y}) \in S_{\text{opt}}(\bar{x})$ by hypothesis. Consequently, $\mathcal{G}$ is locally Lipschitz continuous at $\bar{x}$, and its lower and upper Dini derivatives satisfy the bounds described in (14). Invoking (Cobzaş et al., 2019, Theorem 2.1.6) gives us Lipschitz continuity of $\mathcal{G}|_{\mathbb{B}(\mathcal{M})^n}$, and our proof is complete. $\square$

From this point forward, we shall operate under the blanket assumption that MFCQ holds at every $(\bar{t}, \bar{y}) \in S_{\text{opt}}(\bar{x})$ for all $\bar{x} \in \mathbb{R}^{N \times n}$ — a property that can, in principle, be verified once the functions $\varphi_1, \ldots, \varphi_k$, are specified.

*Remark* 3.3. We highlight that reformulating (1) into (8) using the variable $t$ and representing $\mathcal{M}$ through continuously differentiable functions $\varphi_1, \ldots, \varphi_k$, was crucial for establishing Theorem 3.2. The existing literature does not appear to have employed regularity properties of marginal functions for the design of optimal covering algorithms, which is a point of departure of our contribution.

*Remark* 3.4. Observe that, in principle, every closed subset of $\mathbb{R}^N$ admits a representation as the zero-level set of a smooth function (Calderón and Zygmund, 1961) that is realized as a smooth regularization of the distance-to-the-set function. However, such a function may be difficult to encode in a finitary way for computational purposes. In this light, while our assumption that the set $\mathcal{M}$ is realized as the intersection of zero-sublevel sets of finitely many continuously differentiable functions may appear to be restrictive, it is a reasonably weak assumption, and this family of sets includes sublevel sets of polynomials in particular.

We now turn to the minimization of $\mathcal{G}$. In view of Proposition 2.1, it suffices to restrict the search space to $\mathbb{B}(\mathcal{M})^n$ since all global minimizers of $\mathcal{G}$ are guaranteed to lie within this set. Moreover, by Theorem 3.2, we see that $\mathcal{G}|_{\mathbb{B}(\mathcal{M})^n}$ is $L$-Lipschitz. Now we leverage recent advances in nonconvex and nonsmooth optimization using randomized smoothing by (Lin et al., 2022; Liu et al., 2024) to design our numerical algorithm. Specifically, we adopt an optimality framework based on generalized Goldstein stationary points (Goldstein, 1977) and develop an algorithm that provably converges to such a point based solely on oracle evaluations of $\mathcal{G}$. Furthermore, we establish a polynomial bound on the number of such oracle calls required for convergence. We proceed by laying out a set of definitions. For a detailed exposition, we refer the reader to (Lin et al., 2022).

**Definition 3.5.** Given a point $x \in \mathbb{R}^{N \times n}$ and a direction $v \in \mathbb{R}^{N \times n}$, the *generalized directional derivative* of $\mathcal{G}$ is defined as $D\mathcal{G}(x; v) := \limsup_{y \to x, \, t \downarrow 0} \frac{\mathcal{G}(y+tv) - \mathcal{G}(y)}{t}$. Then, the *generalized gradient* (Clarke, 1990) of $\mathcal{G}$ is defined as the set

$$\partial \mathcal{G}(x) := \left\{ g \in \mathbb{R}^{N \times n} \mid \langle g, v \rangle \leqslant D\mathcal{G}(x; v), \text{ for all } v \in \mathbb{R}^d \right\}. \tag{15}$$

Let $\mathbb{B}(x, \delta) := \{ y \in \mathbb{R}^{N \times n} \mid \|y - x\| \leqslant \delta \}$. Given a $\delta \geqslant 0$, the *Goldstein $\delta$-subdifferential (Goldstein, 1977) of $\mathcal{G}$ at $x$* is defined as $\partial_\delta \mathcal{G}(x) := \text{conv} \left( \bigcup_{y \in B(x, \delta)} \partial \mathcal{G}(y) \right)$. Given a $\delta \geqslant 0$ and $\gamma, \varepsilon > 0$, a point $x \in \mathbb{R}^{N \times n}$ is called a $(\gamma, \delta, \varepsilon)$-*generalized Goldstein stationary point (Liu et al., 2024) of $\mathcal{G}|_{\mathbb{B}(\mathcal{M})^n}$* if $\min \left\{ \frac{1}{\gamma} \left\| x - \text{Proj}_{\mathbb{B}(\mathcal{M})^n}(x - \gamma g) \right\| \mid g \in \partial_\delta \mathcal{G}(x) \right\} \leqslant \varepsilon$, where $\text{Proj}_{\mathbb{B}(\mathcal{M})^n}(y)$ denotes the orthogonal projection of $y$ onto $\mathbb{B}(\mathcal{M})^n$.

We now present our algorithm — an adaptation of techniques from (Lin et al., 2022; Liu et al., 2024) with minor modifications suited to the minimization of $\mathcal{G}$ restricted to $\mathbb{B}(\mathcal{M})^n$.

---

**Algorithm 1** `gradOptNet`: a numerical gradient-based optimal covering algorithm

---

1: **Input:** Initial point $x_0 \in \mathbb{B}(\mathcal{M})^n$, stepsize $\gamma > 0$, smoothing parameter $\delta$, iteration number $T \geqslant 1$, and parameters $b_1, b_2$, and $q$.
2: **for** $t = 0, 1, \ldots, T-1$ **do**
3:     **if** $\text{mod}(t, q) = 0$ **then**
4:         Sample $w_{1,t}, \ldots, w_{b_1,t}$ uniformly from the unit sphere in $\mathbb{R}^{N \times n}$.
5:         Let $g_{i,t} = \frac{Nn}{2\delta} \left( \mathcal{G}(x_t + \delta w_{i,t}) - \mathcal{G}(x_t - \delta w_{i,t}) \right) w_{i,t}$ for each $i \in \{1, \ldots, b_1\}$.
6:         Set $v_t = \frac{1}{b_1} \sum_{i=1}^{b_1} g_{i,t}$.
7:     **else**
8:         Sample $w_{1,t}, \ldots, w_{b_2,t}$ uniformly from the unit sphere in $\mathbb{R}^{N \times n}$.
9:         Let $g_{i,t} = \frac{Nn}{2\delta} \left( \mathcal{G}(x_t + \delta w_{i,t}) - \mathcal{G}(x_t - \delta w_{i,t}) \right) w_{i,t}$ for each $i \in \{1, \ldots, b_2\}$.
10:         Let $g_{i,t-1} = \frac{Nn}{2\delta} \left( \mathcal{G}(x_{t-1} + \delta w_{i,t}) - \mathcal{G}(x_{t-1} - \delta w_{i,t}) \right) w_{i,t}$ for each $i \in \{1, \ldots, b_2\}$.
11:         Set $v_t = \frac{1}{b_2} \sum_{i=1}^{b_2} (g_{i,t} - g_{i,t-1}) + v_{t-1}$.
12:     **end if**
13:     Update $x_{t+1} = \text{Proj}_{\mathbb{B}(\mathcal{M})^n}(x_t - \gamma v_t)$.
14: **end for**
15: **return** $x_R$, where $R \in \{0, 1, \ldots, T-1\}$ is uniformly sampled

---

**Theorem 3.6.** *Consider the problem* (8) *and grant the notations established above. Let $L$ denote the Lipschitz constant of $\mathcal{G}|_{\mathbb{B}(\mathcal{M})^n}$. With $b_1 = \mathcal{O}\left( \frac{NnL^2}{\varepsilon^2} \right)$, $b_2 = q = \mathcal{O}\left( \frac{\sqrt{Nn}L}{\varepsilon} \right)$, and $\gamma = \frac{\delta}{2NnL}$, the Algorithm 1 requires atmost $\mathcal{O}\left( N^{\frac{3}{2}} n^2 \epsilon(\mathcal{M}, 1) L^3 \delta^{-1} \varepsilon^{-3} \right)$ calls of $\mathcal{G}$ to obtain a $(\gamma, \delta, \varepsilon)$-generalized Goldstein stationary point of $\mathcal{G}$ in expectation.*

*Proof.* In view of Theorem 3.2, we have Lipschitz continuity of $\mathcal{G}|_{\mathbb{B}(\mathcal{M})^n}$. We observe that Algorithm 1 corresponds to a variant of (Liu et al., 2024, Algorithm 3), adapted to the setting in which the objective function is deterministic. Noting that the diameter of $\mathbb{B}(\mathcal{M})^n$ is $4\sqrt{n}\,\epsilon(\mathcal{M}, 1)$, we invoke (Liu et al., 2024, Corollary 5.4) to obtain the complexity bound in the assertion. $\qquad \square$

*Remark* 3.7. Theorem 3.2 provided the substrate to enable numerical gradient-based algorithms to solve (1). Since the outer minimization problem in (1) is non-convex, numerical gradient-based

algorithms can only guarantee convergence to stationary points as in 3.6. We attempt to obtain global solutions to (1) in two steps. First, Algorithm 1 involves the calculation of difference quotients and their randomized smoothing; the smoothing is necessary for obtaining 'averaged' descent directions to solve non-convex optimization problems up to stationarity, and this is performed along with a variance reduction scheme as proposed in (Liu et al., 2024). Second, global convergence to minimizers (as opposed to stationary points) is facilitated by the employment of several randomized 'restarts' (parallel initializations) in the indicated local scheme.

## 4 Numerical experiments

This section details the numerical experiments carried out using Algorithm 1. Heuristic observations in (Liu et al., 2024, Section 6) align with our observations: faster convergence is observed when the variance reduction step is omitted (i.e., $q = 1$, rendering $b_2$ immaterial). We also observe better performance when $b_1$ is increased in response to larger values of $n$ or $N$. Accordingly, we set $q = 1$ for all our experiments. For the ease of visualization, we present examples with $N = 2$ and $n < 10$ in §4; an illustration with $N = 3$ is presented in Example D.2 in §D. At each point, $\mathcal{G}$ is evaluated using the NLP solver BARON (Sahinidis, 1996), known for its global optimization capabilities. To the best of our knowledge, no existing benchmarks or ground truth results are available in the literature, and therefore, our comparisons of computation time and accuracy are with techniques like simulated annealing that do not leverage the Lipschitz continuity of $\mathcal{G}$ for outer minimization.

**Example 4.1.** Consider the case where $N = 2$, $k = 1$, and the set $\mathcal{M}$ is the solid ellipse

$$\mathcal{M} := \left\{ y := (y_1, y_2) \in \mathbb{R}^2 \ \middle|\ \varphi_1(y) := \frac{y_1^2}{9} + \frac{y_2^2}{4} - 1 \leqslant 0 \right\}. \tag{16}$$

The application of Algorithm 1 with multiple parallel initializations (to enhance global convergence efficiency), led to the results presented in Fig. 1 for number of balls $n = 2, 3, 4, 5, 7,$ and $8$.

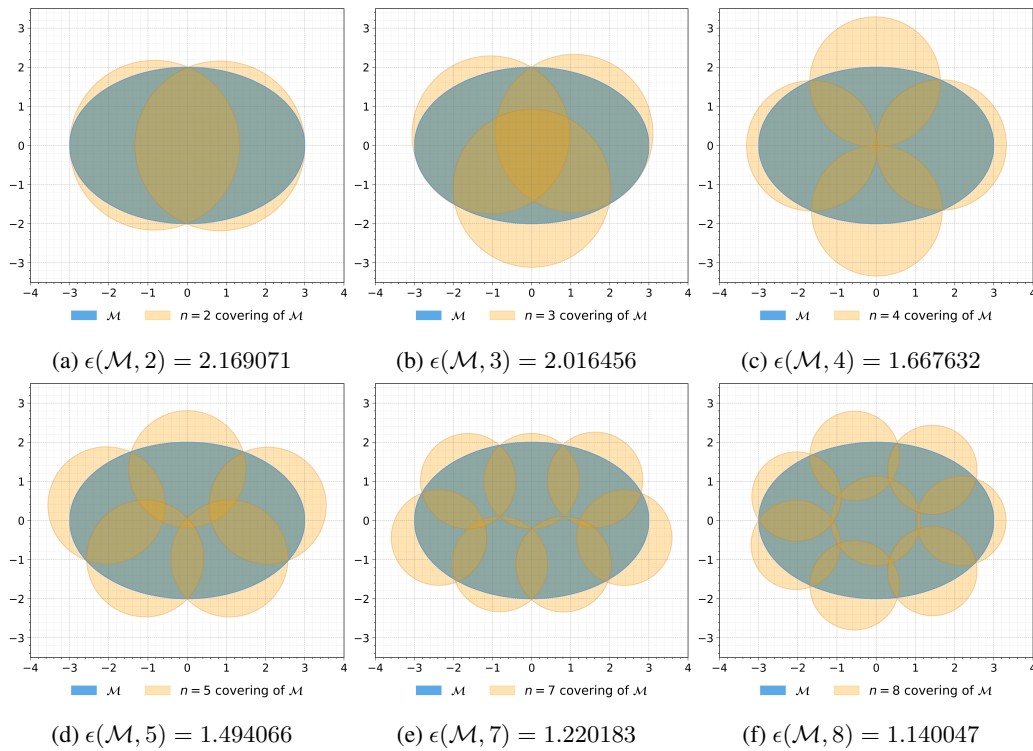

(a) $\epsilon(\mathcal{M}, 2) = 2.169071$     (b) $\epsilon(\mathcal{M}, 3) = 2.016456$     (c) $\epsilon(\mathcal{M}, 4) = 1.667632$

(d) $\epsilon(\mathcal{M}, 5) = 1.494066$     (e) $\epsilon(\mathcal{M}, 7) = 1.220183$     (f) $\epsilon(\mathcal{M}, 8) = 1.140047$

Figure 1: Covering of the ellipse (16) for varying values of $n$.

The results exhibit a visually interpretable structure primarily due to the inherent symmetry of the set $\mathcal{M}$. This naturally motivates the expectation of symmetric structures in the optimal coverings—a property that is indeed observed in the solutions reported in Fig. 1. The evolution of the objective

value for the case $n = 4$ against the iterations of Algorithm 1, with $\gamma = 0.05, \delta = 0.01, b_1 = 24$, and $q = 1$ is shown in Fig. 4 featuring 4 restarts. For each $n$, Algorithm 1 executed with 10 restarts yielded final radius values with relative standard deviation $< 1\%$, indicating excellent robustness. Table 1 tabulates and compares the computation time and solution quality for Algorithm 1 vs Simulated Annealing (Metropolis Hastings sampling and geometric cooling rate) for the example with $n = 4$..

| Method | $\epsilon(\mathcal{M}, 4) \downarrow$ | | | Number of evaluations of $\mathcal{G} \downarrow$ |
| --- | --- | --- | --- | --- |
| | Trial 1 | Trial 2 | Trial 3 | |
| gradOptNet (Ours) | 1.682 | 1.706 | 1.698 | 4800 |
| Simulated Annealing | 2.227 | 2.574 | 2.301 | 10000 |

Table 1: Comparison of computation time and solution quality across methods.

*Remark* 4.2. We emphasize that the ability of the inner solver used to evaluate $\mathcal{G}$ to attain global optimality is crucial for the effectiveness of Algorithm 1. If a solver lacking global optimality guarantees (such as IPOPT) is employed, it may fail to produce a valid covering of $\mathcal{M}$. This phenomenon is treated in Example D.1.

**Example 4.3.** Let us consider an example where $\mathcal{M}$ is asymmetric and nonconvex, and to this end, consider the case where $N = 2$, $k = 4$, and the set $\mathcal{M}$ is the region

$$\mathcal{M} := \left\{ (y_1, y_2) \in \mathbb{R}^2 \mid \varphi_j(y) \leqslant 0 \text{ for } j = 1, 2, 3, 4 \right\} \text{ where, with } y := (y_1, y_2),$$

$$\varphi_1(y) := 2y_1^2 - y_2, \ \varphi_2(y) := y_2 - 2(y_1 - 1)^2, \tag{17}$$

$$\varphi_3(y) := y_2 - 5(y_1 + 0.1)^2, \text{ and } \varphi_4(y) := -0.1 - y_1.$$

Once again, Algorithm 1 was employed with multiple parallel initializations, and the results presented in Fig. 2 were obtained for $n = 1, 2, 3, 4, 5, 6$.

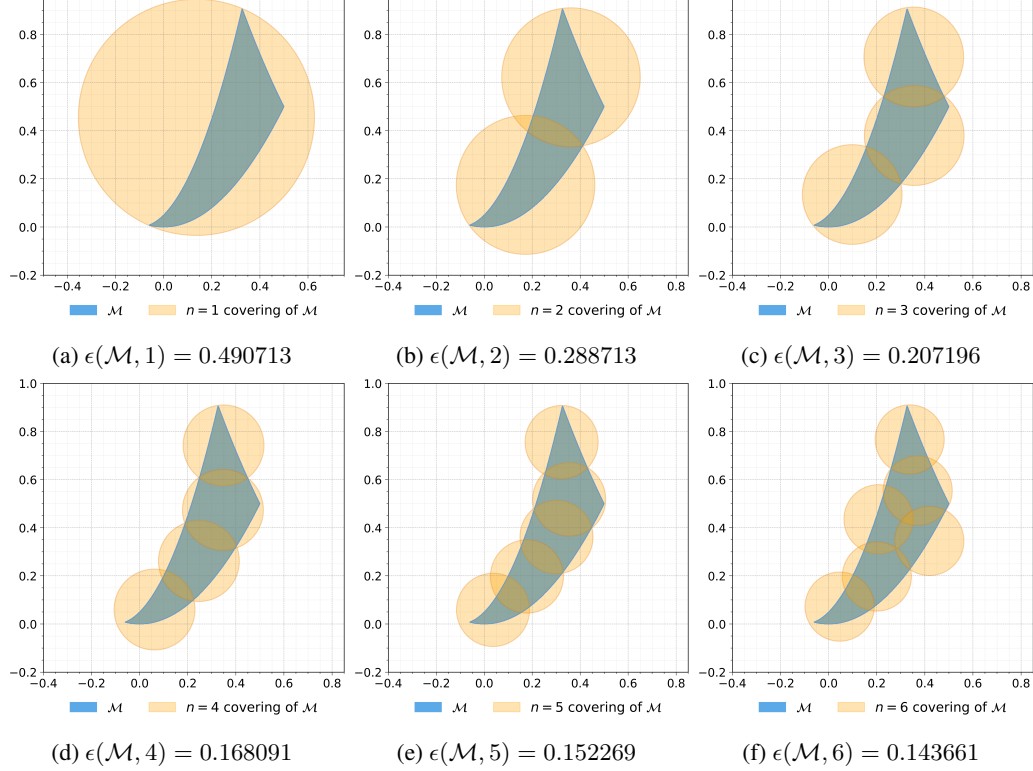

(a) $\epsilon(\mathcal{M}, 1) = 0.490713$    (b) $\epsilon(\mathcal{M}, 2) = 0.288713$    (c) $\epsilon(\mathcal{M}, 3) = 0.207196$

(d) $\epsilon(\mathcal{M}, 4) = 0.168091$    (e) $\epsilon(\mathcal{M}, 5) = 0.152269$    (f) $\epsilon(\mathcal{M}, 6) = 0.143661$

Figure 2: Covering of the region (17) for $n = 1, 2, 3, 4, 5$, and 6.

Similarly, for each $n$, Algorithm 1 executed with 10 restarts converged to final radius values with relative standard deviation $< 1\%$. Additional experimental details are provided in §D.

# 5 Concluding remarks

In this work we established a computationally viable approach to the optimal covering problem. We started with the definition of the problem (1) and applied a sequence of reformulations under mild structural assumptions to derive a version (7) more amenable to analysis. A central contribution of our work is a novel result establishing Lipschitz continuity of the optimal value function $\mathcal{G}$ arising from the nonconvex inner problem, which forms the foundation for applying a numerical gradient-based (zeroth-order) algorithm for the outer minimization. This algorithm leverages recent advances in nonsmooth and nonconvex optimization and, via randomized smoothing, offers non-asymptotic convergence guarantees for obtaining approximate stationary points. We demonstrated the effectiveness of our method through experiments on compact subsets of $\mathbb{R}^2$ and $\mathbb{R}^3$. Future directions include accelerating the computation of $\mathcal{G}$ for convex sets via DC programming techniques and developing strategies to improve global convergence guarantees for the outer minimization. We also observe that $\mathcal{G}(x_1, \ldots, x_n)$ is a permutation invariant function. This property can be used to restrict the search space further, which may help accelerate convergence.

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

## A Foundational Definitions and Results

Here we outline a few important definitions from sensitivity analysis and nonsmooth optimization.

**Definition A.1.** A set-valued mapping $S : A \rightrightarrows B$ is a mapping $S : A \to 2^B$, where for each $x \in A$, $S(x)$ is a subset of $B$. They have been extensively used to study the feasible regions and solution maps of optimization problems

**Definition A.2.** The set valued map $S : A \rightrightarrows B$ is said to be uniformly compact near $\bar{x}$ if the set $\bigcup_{x \in N(\bar{x})} S(x)$ is bounded for some neighborhood $N(\bar{x})$ of $\bar{x}$.

**Definition A.3.** A mapping $f : \mathbb{R}^\nu \to \mathbb{R}$ is said to be locally Lipschitz if at each point $x \in \mathbb{R}^\nu$, there exists a neighborhood $\mathcal{O}$ and $L \geqslant 0$ such that $|f(x_1) - f(x_2)| \leqslant L\|x_1 - x_2\|$ for all $x_1, x_2 \in \mathcal{O}$.

**Definition A.4.** A mapping $f : U \subset \mathbb{R}^\nu \to \mathbb{R}$ is $L$-Lipschitz if there exists $L \geqslant 0$ such that for every $x_1, x_2 \in U$, we have $|f(x_1) - f(x_2)| \leqslant L\|x_1 - x_2\|$. The number $L$ is called the Lipschitz constant of $f$.

**Corollary A.5** ((Cobzaş et al., 2019)). *Let $f : \mathbb{R}^\nu \to \mathbb{R}$ be a locally Lipschitz function and $U \subset \mathbb{R}^\gamma$ be compact. Then the restriction of $f$ to $U$ denoted by $f|_U$ is L-Lipschitz.*

We define Dini-derivatives to generalize the notion of directional differentiability for non-smooth functions.

**Definition A.6.** Let $f : \mathbb{R}^\nu \to \mathbb{R}$ and $x \in \mathbb{R}^\nu$. The upper and lower Dini-derivatives of $f$ along the direction $v \in \mathbb{R}^\nu$ are defined as

$$D^+ f(x; h) := \limsup_{h \downarrow 0} \frac{1}{h}(f(x + hv) - f(x)) \text{ and}$$

$$D^- f(x; h) := \liminf_{h \downarrow 0} \frac{1}{h}(f(x + hv) - f(x)) \tag{18}$$

respectively.

**Definition A.7.** Given a point $x \in \mathbb{R}^\nu$ and a direction $v \in \mathbb{R}^\nu$, the *generalized directional derivative* of a nondifferentiable function $f$ is defined as $Df(x; v) := \limsup_{y \to x, \, t \downarrow 0} \frac{f(y+tv)-f(y)}{t}$. Then, the *generalized gradient* (Clarke, 1990) of $f$ is defined as the set

$$\partial f(x) := \left\{ g \in \mathbb{R}^\nu \mid g^\top v \leqslant D\mathcal{G}(x; v), \text{ for all } v \in \mathbb{R}^d \right\}. \tag{19}$$

**Definition A.8.** Let $\mathbb{B}(x, \delta) := \{y \in \mathbb{R}^\nu \mid \|y - x\| \leqslant \delta\}$. Given a $\delta \geqslant 0$, the *Goldstein $\delta$-subdifferential (Goldstein, 1977) of $f$ at $x$* is defined as

$$\partial_\delta f(x) := \text{conv} \left( \bigcup_{y \in B(x,\delta)} \partial f(y) \right). \tag{20}$$

## B Proofs of Propositions 2.1 and 2.2

*Proposition 2.1.* We know that for every $i \in \{1, \ldots, n\}$ there exists $y \in \mathcal{M}$ such that $\|y - x_i^*\| \leqslant \epsilon(\mathcal{M}, n)$. Consequently, $\|z^* - x_i^*\| \leqslant \|z^* - y\| + \|y - x_i^*\|$, which leads to

$$\|z^* - x_i^*\| \leqslant \epsilon(\mathcal{M}, n) + \epsilon(\mathcal{M}, 1),$$

and since $\epsilon(\mathcal{M}, n') \leqslant \epsilon(\mathcal{M}, n)$ for every $n' > n$,

$$\|z^* - x_i^*\| \leqslant 2\epsilon(\mathcal{M}, 1),$$

and the assertion follows. $\qquad \square$

*Proposition 2.2.* Strict feasibility of both problems is always ensured since any point in $\mathcal{M}$ with $t = 0$ readily satisfies all the constraints. It follows immediately that $(t^\dagger, y^\dagger)$ is a feasible point of (6), which implies $t^\dagger \leqslant t^*$. Moreover, since

$$t^* = \max \left\{ t \in \mathbb{R} \mid t \leqslant \| y^* - x_i \| \text{ for all } i = 1, \ldots, n \right\},$$

it follows that $t^* \leqslant t^\dagger$. Consequently, $t^\dagger = t^*$. □

# C   Lipschitz continuity of optimal value functions of parametrized nonlinear programs

Consider a parametrized nonlinear program of the form

$$\begin{aligned} \min_{z} \quad & g_0(x, z) \\ \text{subject to} \quad & g_i(x, z) \leqslant 0 \text{ for } i = 1, \ldots, p, \end{aligned} \tag{21}$$

where the functions $g_0, g_1, \ldots, g_p$ are continuously differentiable from $\mathbb{R}^\gamma \times \mathbb{R}^\kappa$ to $\mathbb{R}$. The Lagrangian for (21) is given by

$$\mathcal{L}(x, z, \lambda) := g_0(x, z) + \sum_{i=1}^{p} \lambda_i g_i(x, z). \tag{22}$$

The feasible region is denoted by the set-valued map $S_{\text{feas}} : \mathbb{R}^\gamma \rightrightarrows \mathbb{R}^\kappa$,

$$S_{\text{feas}}(\bar{x}) := \left\{ z \in \mathbb{R}^\kappa \mid g_i(\bar{x}, z) \leqslant 0 \text{ for } i = 1, \ldots, p \right\}. \tag{23}$$

The optimal value function $\Phi : \mathbb{R}^\gamma \to \mathbb{R} \cup \{+\infty\}$ is the optimal value of the NLP for a given parameter $\bar{x} \in \mathbb{R}^\gamma$. That is,

$$\Phi(\bar{x}) := \min_{z \in S_{\text{feas}}(\bar{x})} g_0(\bar{x}, z). \tag{24}$$

By convention, $\Phi(\bar{x}) = +\infty$ if the problem is infeasible. The set of global optimizers is the set-valued map $S_{\text{opt}} : \mathbb{R}^\kappa \rightrightarrows \mathbb{R}^\gamma$,

$$S_{\text{opt}}(\bar{x}) := \left\{ z \in S_{\text{feas}}(\bar{x}) \mid g_0(\bar{x}, z) = \Phi(\bar{x}) \right\}. \tag{25}$$

For a choice $\bar{x}$ of $x$, let $\bar{z} \in S_{\text{opt}}(\bar{x})$. We define the set of active constraint indices at $(\bar{x}, \bar{z})$ as

$$I(\bar{x}, \bar{z}) := \left\{ i \in \{1, \ldots, p\} \mid g_i(\bar{x}, \bar{z}) = 0 \right\}. \tag{26}$$

The set of Karush-Kuhn-Tucker (KKT) vectors of (8) for a pair $(\bar{x}, \bar{z}) \in \mathbb{R}^\gamma \times \mathbb{R}^\kappa$ is denoted by

$$\Lambda(\bar{x}, \bar{z}) := \{\lambda \in [0, +\infty[^p \mid \nabla_z \mathcal{L}(\bar{x}, \bar{z}, \lambda) = 0, \lambda_i g_i(\bar{x}, \bar{z}) = 0 \text{ for } i = 1, \ldots, p\} \tag{27}$$

The *Mangasarian-Fromovitz constraint qualification (MFCQ)* is said to hold at a point $\bar{z} \in S_{\text{opt}}(\bar{x})$ if there exists a direction $w \in \mathbb{R}^\kappa$ such that the inner product of $w$ with the gradient of each active constraint is strictly negative. To wit, the following condition must hold:

$$\nabla_z g_i(\bar{x}, \bar{z})^\top w < 0 \text{ for all } i \in I(\bar{x}, \bar{z}) \tag{28}$$

We now state the result (Fiacco and Ishizuka, 1990, Theorem 4.2) that establishes the conditions under which the optimal value function is Lipschitz continuous.

**Theorem C.1.** *Consider the problem* (21) *and grant the notations established above. Fix $\bar{x} \in \mathbb{R}^\gamma$ and suppose that*

(C.1-a) *the MFCQ condition holds for every $\bar{z} \in S_{\text{opt}}(\bar{x})$, and*

(C.1-b) *the map $S_{\text{feas}}$ is uniformly compact near $\bar{x}$.*

*Then the function $\Phi$ is locally Lipschitz continuous at $\bar{x}$. Moreover, for any $v \in \mathbb{R}^\gamma$, we have*

$$\inf_{\bar{z} \in S_{\text{opt}}(\bar{x})} \min_{\lambda \in \Lambda(\bar{x}, \bar{z})} \frac{\partial \mathcal{L}}{\partial x}(\bar{x}, \bar{z}, \lambda) \cdot v \leqslant -\liminf_{h \downarrow 0} \frac{1}{h}(\Phi(\bar{x} + hv) - \Phi(\bar{x}))$$

$$\leqslant -\limsup_{h \downarrow 0} \frac{1}{h}(\Phi(\bar{x} + hv) - \Phi(\bar{x})) \leqslant \inf_{\bar{z} \in S_{\text{opt}}(\bar{x})} \max_{\lambda \in \Lambda(\bar{x}, \bar{z})} \frac{\partial \mathcal{L}}{\partial x}(\bar{x}, \bar{z}, \lambda) \cdot v.$$

# D Experimental details

In this section, we provide further details of the experimental procedure along with a 3D illustration of covering in Example D.2. As emphasized in §1, the main computational bottleneck arises from having to reliably solve the inner problem (8) for the evaluation of $\mathcal{G}$ globally. This is the reason a solver with global optimality guarantees is employed. To illustrate the phenomenon mentioned in 4.2, we provide the following example.

**Example D.1.** For the solid ellipse (16) in Example 4.1. Here we employed Algorithm 1 maintaining all parameters from Example 4.1, except that $\mathcal{G}$ was evaluated using `IPOPT` instead of `BARON` in Algorithm 1. Fig. 3 illustrates the failure of Algorithm 1 in producing an adequate cover of $\mathcal{M}$ with $n = 1, 2$, and 3, when `IPOPT` was employed.

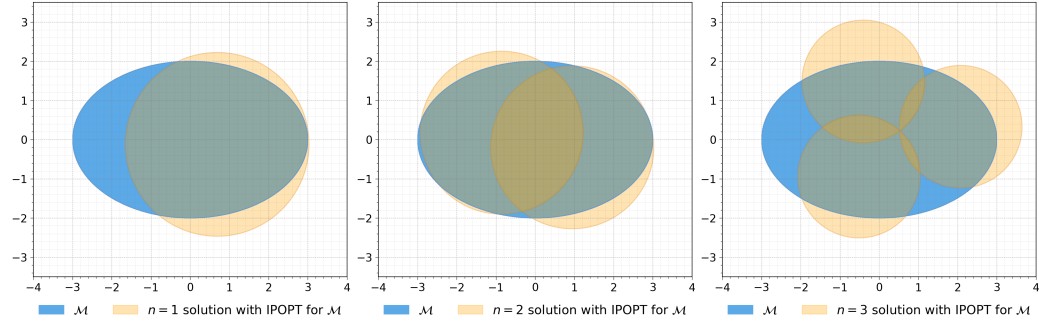

Figure 3: Covering fails if `IPOPT` is employed in place of `BARON` for $n = 1, 2, 3$.

For our experiments, we used the demo mode of `BARON`, which imposes limits on the problem dimensions and number of constraints. To facilitate faster convergence and support for larger-scale problems, a more advanced license may be used. For more details, visit `https://minlp.com/baron-licenses`. Table 2 reports the time taken to compute $\mathcal{G}$ in Example 4.1 for varying $n$ using `BARON` in demo mode on an AMD Ryzen 7 4800H 2.90 GHz CPU. Note that the total number of

Table 2: Computation time for $\mathcal{G}$ for varying $n$

| $n$ | Computation time (ms) |
|---|---|
| 2 | 129 |
| 3 | 135 |
| 4 | 143 |
| 5 | 146 |
| 6 | 154 |
| 7 | 160 |
| 8 | 164 |
| 9 | 173 |

evaluations of $\mathcal{G}$ per iteration of Algorithm 1 depends on $b_1$ and $b_2$.

Fig 4 shows the temporal evolution of the objective in Example 4.1 and features 4 restarts (initializations). An example with a nonconvex, asymmetric set in $\mathbb{R}^3$ is provided in Example D.2. Note that only the isometric view is provided for each case.

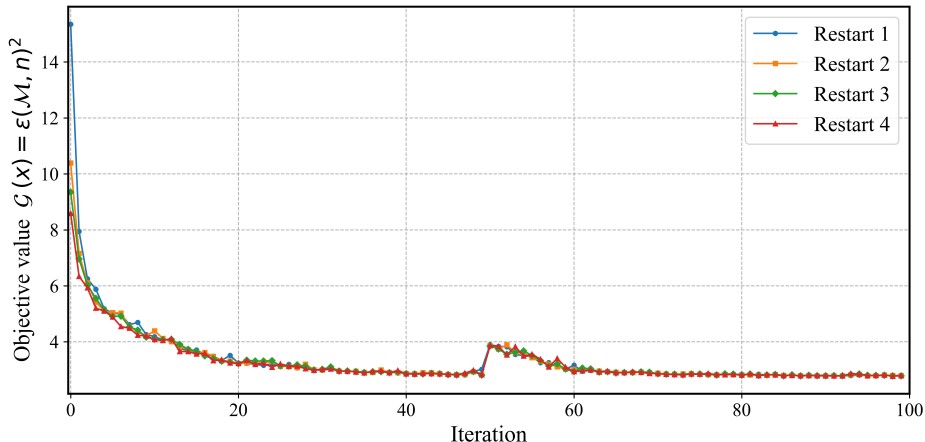

Figure 4: The evolution of $\mathcal{G}$ against iteration count for the problem in Example 4.1 for $n = 4$.

**Example D.2.** As an illustration of the 3D case, we present an example where $\mathcal{M}$ is an asymmetric and nonconvex subset of $\mathbb{R}^3$. Consider the case where $k = 3$, and the set $\mathcal{M}$ is the region

$$\mathcal{M} := \left\{ (y_1, y_2, y_3) \in \mathbb{R}^3 \mid \varphi_j(y) \leqslant 0 \text{ for } j = 1, 2, 3 \right\} \text{ where, with } y := (y_1, y_2, y_3),$$

$$\varphi_1(y) := y_3 - (1 - y_1^2 - 0.3y_2^2), \ \varphi_2(y) := -(y_3 + 1), \text{ and } \varphi_3(y) := y_2 - (0.5(y_1 - 1)^2 + 0.2y_3^2). \tag{29}$$

Algorithm 1 was employed with multiple restarts, and pictures of the outcomes are presented in Fig. 5 for the cases $n = 1, 2, 3,$ and $4$.

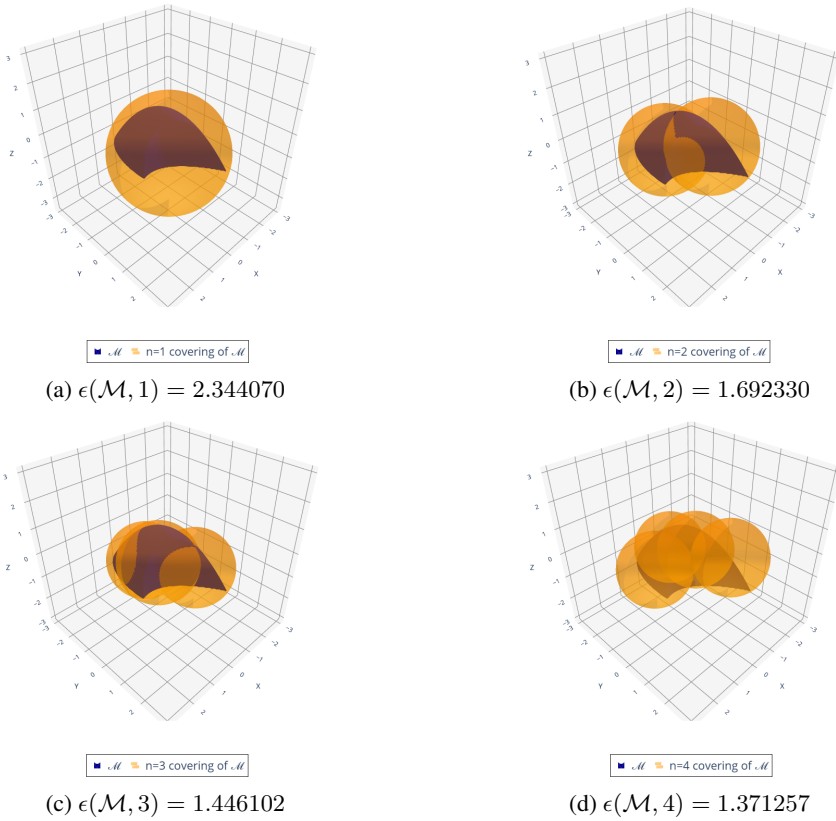

(a) $\epsilon(\mathcal{M}, 1) = 2.344070$

(b) $\epsilon(\mathcal{M}, 2) = 1.692330$

(c) $\epsilon(\mathcal{M}, 3) = 1.446102$

(d) $\epsilon(\mathcal{M}, 4) = 1.371257$

Figure 5: Covering of the region (29) for $n = 1, 2, 3,$ and $4$.

