# OpenReview forum: "A Computationally Viable Numerical Gradient-based Technique for Optimal Covering Problems"
_NeurIPS.cc/2025/Conference — NeurIPS 2025 poster_

### Official Review · Reviewer_xqmn · 2025-06-11

**Clarity:** 3
**Significance:** 2
**Originality:** 3
**Rating:** 5
**Confidence:** 2

**Summary:**

The paper studies the optimal covering problem, where the goal is to obtain the optimal way to cover a compact subset of Euclidean space with a given number of Euclidean balls, where the optimality is formulated based on the maximum distances of the centers of the balls to the subset. The paper proposes a reformulation of the original problem to make it possess desirable regularity properties, and thus propose an algorithm based on numerical gradient to solve the problem. The efficiency of the algorithm is quantified, and numerical experiments are provided. The experiments demonstrate that the proposed algorithm indeed achieves good performances.

**Questions:**

I am not that familiar with the tackled problem. As briefly mentioned in Strengths and Weaknesses, I find the empirical results relatively weak. To me the presented empirical results largely show that the algorithm was able to find reasonable results, without showing that it finds really good results or has good convergence properties due to the lack of a ground truth solution or relevant baselines. While the authors argued that no competing algorithms are available in the literature in Section 4, the authors also mentioned some previous approaches in Section 1. It would be ideal if the authors can improve the part on numerical results using as baselines some previous approaches, thus providing empirical evidence that the proposed algorithm is indeed a significant improvement upon the previous ones either in terms of computational efficiency or accuracy or both.

**Ethical Concerns:**

["NO or VERY MINOR ethics concerns only"]

**Final Justification:**

My sole concern was about the lack of a ground truth solution or a relevant baseline, thus limiting the empirical evidence. The authors did provided empirical baselines that demonstrated the good properties of the proposed algorithm. As such, I raise my score to 5.

**Limitations:**

yes

**Paper Formatting Concerns:**

No major concerns.

**Quality:**

3

**Strengths And Weaknesses:**

Strengths:

1. Sufficient background is provided, motivating the research problem and demonstrating its importance.

2. The proposed algorithm is theoretically grounded, and is proven to yield reasonable performances.

Weaknesses:

1. As acknowledged by the authors, the proposed algorithm is still computationally costly.

2. The empirical results largely serve to demonstrate the correctness of the proposed algorithm, without showing its efficiency.

---

> ### Author Rebuttal · Authors · 2025-07-28
>
> # Strength:
>
> We thank the reviewer for their positive comments regarding the quality of the paper.
>
>
> # Questions:
>
> > Q1: To me the presented empirical results largely show that the algorithm was able to find reasonable results, without showing that it finds really good results or has good convergence properties due to the lack of a ground truth solution or relevant baselines.
>
> The problem tackled in our submission is quite difficult; optimal solutions in only a few isolated cases with excellent symmetry are known today. We set out to establish the algorithmic framework contained in our submission with solid theoretical backing from Theorems 3.2 and 3.6, and that constitutes our primary contribution; the numerical implementations were presented as proof-of-concept. Please note that Theorem 3.6 provides guarantees in expectation of the convergence of the outer problem in Theorem 3.6 by leveraging our Lipschitz continuity result in Theorem 3.2 along with the recent developments made in nonsmooth nonconvex optimization in [1]. An alternative generalized framework for obtaining solutions for the same has not appeared in the literature. We acknowledge that indeed, our experimental results are illustrations and are currently outside the scope of our submission.
>
> > Q2: While the authors argued that no competing algorithms are available in the literature in Section 4, the authors also mentioned some previous approaches in Section 1. It would be ideal if the authors can improve the part on numerical results using as baselines some previous approaches, thus providing empirical evidence that the proposed algorithm is indeed a significant improvement upon the previous ones either in terms of computational efficiency or accuracy or both.
>
> To the best of our knowledge, no other algorithm gives theoretical guarantees for optimality for the problem under consideration. It is well-known to be hard, even in the $n = 1$ case -- this is the Chebyshev center problem. However, it is possible to compare our numerical gradient-based outer minimization that exploits the Lipschitz continuity of $\mathcal{G}$ given in Theorem 3.2 against other techniques like simulated annealing that do not leverage this regularity. For this, we present below the same experimental result in our reply to Reviewer 3wcR (Reviewer #2).
>
> Here we tabulate and compare the computation time and solution quality for Algorithm 1 vs Simulated Annealing for the 2D ellipse example with $n=4$, both on an AMD Ryzen 7 4800H CPU. Due to demo license constraints of the proprietary solver BARON, cluster-scale parallelization was not leveraged for the initial submission.
>
> | Method              | Covering radius obtained (Trial 1) | Covering radius obtained (Trial 2) | Covering radius obtained (Trial 3) | Number of evaluations  of inner problem |
> |---------------------|:--------------------------:|:--------------------------:|:-------------------------:|:---------------------------------------:|
> |  Algorithm 1 (Ours) |            1.682           |            1.706           |           1.698           |                   4800                  |
> | Simulated Annealing |            2.227           |            2.574           |           2.301           |                  10000                  |
>
> We see from the table that for the same initializations, simulated annealing -- a method that does not leverage regularity properties of $\mathcal{G}$ -- requires more calls of $\mathcal{G}$ to attain convergence and is relatively suboptimal compared to our method. If accepted, we will add this comparative study to the revision.
>
> [1] Z. Liu, C. Chen, L. Luo, and B. K. H. Low. Zeroth-order methods for constrained nonconvex nonsmooth stochastic optimization. In Proceedings of the 41st International Conference on Machine Learning, volume 235, pages 30842–30872, 2024.

---

> > ### Comment · Reviewer_xqmn · 2025-08-03
> >
> > I thank the authors for providing new experimental results, which improved the empirical analysis. I will reconsider my score.

---

> > > ### Author Response · Authors · 2025-08-04
> > > **Re:**
> > >
> > > Thank you for your positive response. We will be happy to address any further questions during the discussion phase.

---

### Official Review · Reviewer_7X5g · 2025-07-03

**Clarity:** 4
**Significance:** 3
**Originality:** 4
**Rating:** 5
**Confidence:** 4

**Summary:**

This paper considers the problem of covering a (possibly nonconvex) bounded set using fixed number, say $n$ balls of smallest possible radius. The quantity of interest are centers of those $n$ balls (called as Chebyshev net of size $n$) and their radius. The main contribution is to reformulate the problem into a  min-max type problem where the inner maximization is nonconcave. The resulting outer value function is nonconcave, nonsmooth and Clarke ``irregular" in variables that represent the location of ball-centers. They show that under mild assumption, the outer Clarke irregular function is Lipschitz continuous. Hence, they can apply zeroth-order methods developed in the recent literature for such nonconvex nonsmooth problems.

**Questions:**

It is necessary for the convergence that the domain of $x$ remains bounded. It is proved that it suffices to search on a bounded space $\mathbb{B}(\mathcal{M})$ which is based on Chebyshev center and radius for case of $n =1$. However, not much is said about the computation of these. Can you expand on how to find such a point?

In line 60, you mentioned a second version in equation (2) where the $n$ centers must belong $\mathcal{M}$ itself. However, not much is said about this problem later on. How would Algorithm 1 change is this problem needs to be solved. Does that also mean projection is required onto set $\mathcal{M}^n$ instead of line 13 of Algorithm 1. Moreover, if $\mathcal{M}$ is nonconvex, does the convergence analysis borrowed from Liu et al. 2024 still hold if one needs to project onto a nonconvex set?

Theorem 3.6 proves (or borrows) the convergence result for $\min_x \mathcal{G}$ problem. However, it does not explicitly state the impact of $b_1, b_2$. Although, it is clear that bigger $b_1,b_2$ will give more smoothing and possibly better estimates of descent direction. Also, I am wondering what are the conditions required on step-size $\gamma$. Usually that is critical to get any convergence/complexity guarantees. Without it, the development looks incomplete to me.

I see that you are treating the variable $x =(x_1, \dots, x_n)$ as a full vector in dimension $Nn$. However, due to the obvious structure of the problem, it probably makes sense to consider block-wise updates and sample $b_1/n$ in unit spheres in $\mathbb{R}^N$, a total of $n$-times. Would that benefit in scaling this approach?

**Ethical Concerns:**

["NO or VERY MINOR ethics concerns only"]

**Final Justification:**

The remarks addressed my concerns.

**Limitations:**

I think the authors clearly mentioned some of the limitations, as those in experiment on Example D.1 where they use IPOPT instead of ``global optimizer" BARON to solve the inner maximization problem.

**Paper Formatting Concerns:**

No.

**Quality:**

3

**Strengths And Weaknesses:**

The paper makes minimal assumptions about the structure of the set $\mathcal{M}$ that needs to be covered. They provide novel insights on how one can prove Lipschitz continuity outer value function $\mathcal{G}(x_1, \dots, x_n)$ using such representation, along with classical results from analysis. This may be of independent interest in other applications. The paper is written with great clarity. I didn't see any obvious typos. All the relevant points are discussed in some detail along with the analysis.

On the weakness side: The formulation requires solving nonconvex maximization to evaluate value function $\mathcal{G}(x)$. Indeed we have $\mathcal{G}(x) = \max t$ s.t. $h(x, y, t) \le 0$ where $h$ is  nonconcave in $y$. One requires global solution of this problem in order to evaluate $\mathcal{G}$ accurately. This is required since zero-th order schemes for problem $\min_x \mathcal{G}(x)$ with randomized smoothing requires high quality estimate to estimate numerical "average" descent directions. Without the global solves of such problem, the method may fail as illustrated in their own experiment in appendix. The good thing is reader is well-cautioned about it in the main paper.
Another weakness is that the algorithmic contribution in this paper are minimal. They are used as blackbox. Some important details and considerations for convergence of these algorithms are missing as well. I have some questions in this regard below.

---

> ### Author Rebuttal · Authors · 2025-07-26
>
> # Strength:
> We thank the reviewer for their positive comments on the paper’s clarity and theoretical insights.
>
> # Weakness:
> We thank the reviewer for their constructive feedback.
> >One requires global solution of this problem in order to evaluate $\mathcal{G}$ accurately. This is required since zero-th order schemes for problem with randomized smoothing requires high quality estimate to estimate numerical "average" descent directions.
>
> The reviewer is correct about the evaluation of $\mathcal{G}$ being computationally costly and Algorithm 1 being reliant on the quality of the solver for the 'inner' optimization problem in (7). However, notice that the problem is intrinsically difficult (even the $n = 1$ case is NP-hard), and consequently this computational expense is not unexpected.
>
> > Another weakness is that the algorithmic contribution in this paper are minimal. They are used as blackbox. Some important details and considerations for convergence of these algorithms are missing as well.
>
> We agree with the reviewer's comments regarding our algorithmic contributions --- the framework backed by the theoretical results Theorem 3.2 and Theorem 3.6 is indeed our key contribution. Theorem 3.2 establishes the regularity property of $\mathcal G$ that is needed for Theorem 3.6 to operate and solve the problem. We leveraged recently established algorithms for nonsmooth, nonconvex optimization for computational purposes. However, once our framework is established and as future algorithms improve complexity bounds under similar assumptions, they can be integrated directly into our problem to obtain optimal covers more efficiently.
>
> # Questions:
> > Q1: It is necessary for the convergence that the domain of $x$ remains bounded...
>
> Indeed, we restrict ourselves to a bounded decision space that depends on the Chebyshev center and radius. A (in fact, the only) numerically viable algorithm for computing these quantities is provided in [1]. However, in our experiments, initializing the centers inside $\mathcal{M}$ and tuning step size $\gamma$ ensured that the iterates of $x$ remained within $\mathbb{B}(\mathcal{M})^n$ (i.e the projection step was bypassed).
>
> > Q2: In line 60, you mentioned a second version in equation (2) where the centers must belong $\mathcal{M}$ itself...
>
> We appreciate the reviewer's question. For convex $\mathcal{M}$, projection onto $\mathcal{M}^n$ would be the only change required in Algorithm 1. The case of nonconvex $\mathcal{M}$ is a scope for future work since it would be a nonsmooth, nonconvex minimization over a nonconvex set and this complicates the projection. If accepted, we will mention that convexity of $\mathcal{M}$ is required to solve the version in equation (2) using the same algorithm with minimal changes.
>
> > Q3: Theorem 3.6 proves (or borrows) the convergence result for...
>
> Theorem 3.6 adapts Corollary 5.4 of [2] with certain modifications suited to the problem. We refer the reviewer to Theorem 5.3 of [2] for the sought result concerning the expected value of the norm of the generalized gradient and its dependence on parameters $\gamma, b_1$, and $b_2$. The same modifications will make the result apply to our problem. We will add this result for completeness in the revision.
>
> > Q4: I see that you are treating the variable $x=(x_1,\ldots,x_n)$ as a full vector in dimension $Nn$...
>
> Our response here is based on what we currently understand from the question: Since the number of decision variables does not change under this alteration, we do not foresee immediate benefits to operating with matrices for improving scalability. Please let us know if you are referring to the fact that the function $\mathcal{G}(x_1,\ldots,x_n)$ is permutation invariant with respect to its input: we seek the reviewer's elaboration on how the proposed treatment might benefit us.
>
>
> [1] P. Paruchuri and D. Chatterjee. Attaining the Chebyshev bound for optimal learning: A numerical algorithm. Systems & Control Letters, 181:105648, 2023.
>
> [2] Z. Liu, C. Chen, L. Luo, and B. K. H. Low. Zeroth-order methods for constrained nonconvex nonsmooth stochastic optimization. In Proceedings of the 41st International Conference on Machine Learning, volume 235, pages 30842–30872, 2024.

---

### Official Review · Reviewer_3wcR · 2025-07-03

**Clarity:** 4
**Significance:** 3
**Originality:** 3
**Rating:** 4
**Confidence:** 2

**Summary:**

This paper considers an optimally covering problem, which is generally NP-hard and computationally challenging. To challenge this problem, this paper proposes a new approach which consists of (according to the authors claim in the paper) (i) A new reformulation of the problem which realizes a smooth objective function, (ii) a new gradient-based algorithm is proposed to minimize the objective function in (i), and its computational complexity is analyzed, and (iii) numerical experiments verifies the actual efficiency of the proposed framework.

**Questions:**

I would like to ask the authors how they think about my suggestion written in Weakness.

From the reasons I wrote above (including the fact that I am not an expert), I begin with a neutral rate. I would like to reconsider my rating depending on the additional information in the discussion stages.

**Ethical Concerns:**

["NO or VERY MINOR ethics concerns only"]

**Limitations:**

The authors do not explicitly state any limitations.
I did not notice any other limitations.

**Quality:**

3

**Strengths And Weaknesses:**

First of all, I have to confess that I am not an expert in this field, and I may be misunderstanding something.  I also have to say that I did not necessarily get into the mathematical detail of the theorems.

Strength: As far as I went through from the abstract to the conclusions, I felt that this paper is very well-organized (well-written). Although I myself is not an expert on this topic, the introduction is clearly written such that the situation in the research field becomes visible. If we completely believe the authors' claim on their contributions, the algorithm is new, which is based on a new reformulation (and thus the development of the new algorithm is not straightforward), and the algorithm is good both theoretically and numerically.

Weakness: Since I could not get into the technical detail, I cannot think of any defects. One thing I noticed is that no information on the actual computational time in the numerical experiments is given. The reason is clarified in the beginning of the numerical experiments section, where it is claimed that no direct comparison of computation time is feasible (since there is no competing existing algorithm). Still, I think some information on actual computation time might be informative for those who hope to employ their algorithm.

---

> ### Author Rebuttal · Authors · 2025-07-26
>
> # Strength:
> We thank the reviewer for their positive comments regarding the clarity of the paper.
> > If we completely believe the authors' claim on their contributions, the algorithm is new, which is based on a new reformulation (and thus the development of the new algorithm is not straightforward), and the algorithm is good both theoretically and numerically.
>
> To the best of our knowledge, the reformulation established in our submission and algorithm are indeed novel, they are supported by strong theoretical guarantees, which are the key contributions of our work. Even the $n = 1$ case of this problem is NP-hard, and there is currently only one algorithm to compute such *optimal* solutions in a finitary way [1]. Once our framework is established, as and when new and improved global optimization solvers are developed, they can be leveraged to obtain optimal covers more efficiently.
>
> # Weakness:
>
> > W1. Some information on actual computation time might be informative for those who hope to employ their algorithm.
>
> We agree that the inclusion of details of actual compute time would benefit the paper. Table 1 in the paper reports per-point computation times for $\mathcal{G}$ on an AMD Ryzen 7 4800H CPU. Since $q=1$ in our experiments, the time taken for Algorithm 1 roughly scales as $2b_1sT$, where $s$ is the per-point computation time. Please note that lines 5,9, and 10 of Algorithm 1 are parallelizable as computation for each $i$ is independent, reducing the time to $2sT$. Due to demo license constraints of the proprietary solver BARON, cluster-scale parallelization was not leveraged for the initial submission. If accepted, we would like to tabulate the computation time details along with a comparison against a gradient-free method like simulated annealing for the outer problem after leveraging the parallelization capabilities of Algorithm 1.
>
> # Experimental Result:
>
> Here we tabulate and compare the computation time and solution quality for Algorithm 1 vs Simulated Annealing for the 2D ellipse example with $n=4$, both on an AMD Ryzen 7 4800H CPU without leveraging the parallelizability.
>
> | Method              | Covering radius obtained (Trial 1) | Covering radius obtained (Trial 2) | Covering radius obtained(Trial 3) | Number of evaluations  of inner problem |
> |---------------------|:--------------------------:|:--------------------------:|:-------------------------:|:---------------------------------------:|
> |  Algorithm 1 (Ours) |            1.682           |            1.706           |           1.698           |                   4800                  |
> | Simulated Annealing |            2.227           |            2.574           |           2.301           |                  10000                  |
>
> We see from the table that for the same initializations, simulated annealing - a method that doesn't leverage regularity properties of $\mathcal{G}$ requires more calls of $\mathcal{G}$ to attain convergence and is relatively suboptimal compared to our method. If accepted, we will add this comparative study to the revision.
>
> [1] P. Paruchuri and D. Chatterjee. Attaining the Chebyshev bound for optimal learning: A numerical algorithm. Systems & Control Letters, 181:105648, 2023.

---

> > ### Comment · Reviewer_3wcR · 2025-08-05
> >
> > Thank you for the numerical data, which should be very helpful for the readers.
> > I would like to keep my score.

---

> > > ### Author Response · Authors · 2025-08-05
> > > **Re: Official Comment by Reviewer 3wcR**
> > >
> > > Thank you for your response. We will be available through the discussion phase should any further point needs to be addressed.

---

### Official Review · Reviewer_Ecfy · 2025-07-18

**Clarity:** 4
**Significance:** 3
**Originality:** 3
**Rating:** 4
**Confidence:** 3

**Summary:**

The manuscript describes an algorithm to construct a Chebyshev net
with $n$ balls for a point set in $\mathbb{R}^N$. Formally, the
optimization problem for the optimal cover amounts to a min max min
problem. The authors suggest a convex relaxation of the inner
minimization in the spirit of quadratic inequality constraints.

The central theorem 3.2 establishes regularity properties that justify
the problem relaxation with Lagrangian (9).
The mansucript presents simple examples of covering an ellipse and a
2-dim. non-convex shape.

**Questions:**

It would be good to receive some
examples of coverings in high dimensions ($N \approx 100$) with 1000
to 10000 balls. How does the algorithm scale in this parameter range.
One might consgtruct concrete covers by finite size scaling in such a
parameter range and this would be a real application with relevance
for ML.

The authors should say something about the scaling in high dimensions,
i.e., point clouds in N dimensions like in coding theory.

**Ethical Concerns:**

["NO or VERY MINOR ethics concerns only"]

**Final Justification:**

For me the strong focus of the paper on optimization is a critical issue from a ML viewpoint.
I lowered my score to borderline accept because I was not satisfied with the explanation of the 10 variable limit in the experiments.
Also coverings in statistical learning are considered in high dimensions.
Both reasons indicate that the manuscript summarizes a somewhat premature research result.

If space permits acceptance then I would support it.

**Limitations:**

yes

**Quality:**

3

**Strengths And Weaknesses:**

Strengths:

The corage problem plays an important role in statistical learning
theory (usually expnential covers) as well as in computational
geometry. The manuscript has a distinct computational geometry flavor
but it might be relevant for approximations in machine learning.

Theoretically the manuscript describes the algorithm quite well. It is
mathematically solid as far as I can judge.

The algorithm is demonstrated by constructing two simple coverings. Scaling

Weaknesses:

I find the topic not central to machine learning or neural
computation. It provides theoretical concepts and an algorithm which
has a weak relation to approximation theory, e.g., \epsilon covers.

The numerical demonstrations are only illustrations but do not refer
to reaL computational problems in ML. The examples in two dimensions
with 8 or 6 covers might not even be interesting for computational
geometry.

---

> ### Author Rebuttal · Authors · 2025-07-28
>
> # Strength:
>
> We thank the reviewer for their positive comments regarding the paper's rigor and clarity.
>
> # Weakness:
>
> > I find the topic not central to machine learning or neural computation. It provides theoretical concepts and an algorithm which has a weak relation to approximation theory, e.g., \epsilon covers.
>
> We acknowledge the reviewer's concern regarding the centrality of the topic to neural computation and machine learning. Though perhaps not central to conventional machine learning, we believe our contributions are well situated in the topic of optimization that features prominently in the CfP of NeurIPS 2025.
>
> > The numerical demonstrations are only illustrations but do not refer to real computational problems in ML. The examples in two dimensions with 8 or 6 covers might not even be interesting for computational geometry.
>
> We agree with the reviewer's comments that the examples provided in numerical experiments appear illustrative. We provide our reasons for this situation in our answer to Q1 below.
>
> # Questions:
>
> > Q1: It would be good to receive some examples of coverings in high dimensions ($N \approx 100$) with 1000 to 10000 balls. How does the algorithm scale in this parameter range. One might construct concrete covers by finite size scaling in such a parameter range and this would be a real application with relevance for ML.
>
> We appreciate the reviewer's question regarding scaling in higher dimensions. For our submission, we could only provide low-dimensional illustrations due to limitations of the proprietary solver BARON (the solver for the inner problem). According to the MINLP website, when BARON runs in demo mode it:
>
>  * Supports up to 10 variables and constraints
>  * Allows 50 nonlinear operations
>
> Hence, it would not be feasible for us to demonstrate higher-dimensional examples in this rebuttal period. Having said that, please note that our key contribution is the algorithmic framework supported by Theorems 3.2 and 3.6, with the examples as proof-of-concept. We have shared our code, which will be open-source, enabling high-dimensional tests with an appropriate BARON license.
>
>
> > Q2: The authors should say something about the scaling in high dimensions, i.e., point clouds in N dimensions like in coding theory.
>
> Theorem 3.6 provides an expected convergence guarantee for the outer minimization in terms of the number of calls of $\mathcal{G}$. It scales with $N$ and $n$ as $\mathcal{O}(N^{\frac{3}{2}}n^2)$ and this is independent of their ranges. However, the inner problem remains nonconvex even after reformulation, and the hardness of solving it globally is inescapable. We have illustrated the failure of a subpar solver for the same as elucidated in Reviewer 7X5g's comments in the weakness section.

---

> > ### Comment · Reviewer_Ecfy · 2025-08-05
> >
> > I see the tendency in ML to focus on optimization (possibly "featuring prominently in the CfP of NeurIPS 2025"). From a conceptual viewpoint, ML and i.e., deep learning should be focussed on controlled approximations. Since we care for generalization and not for empirical risk minimization, the optimization aspect of ML is at best a proxy answer with an often poorly justified regularization attempt.
> > That is the reason why I did not see the manuscript centrally anchored in the core focus of NeurIPS.
> >
> > The answer to Q2 again points to my above argument: Why should I care about convergence of the outer minimization when we aim for good generalization on future data? Generalization is particularly critical for high dimensions as sparsity in high dimensional statistics has shown in the last three decades.
> >
> > Q1: The authors' answer indicates that the research is promising but somewhat premature for a presentation at NeurIPS '25.

---

> > > ### Author Response · Authors · 2025-08-07
> > > **Re: Official Comment by Reviewer Ecfy**
> > >
> > > > I see the tendency in ML to focus on optimization (possibly "featuring prominently in the CfP of NeurIPS 2025"). From a conceptual viewpoint, ML and i.e., deep learning should be focussed on controlled approximations. Since we care for generalization and not for empirical risk minimization, the optimization aspect of ML is at best a proxy answer with an often poorly justified regularization attempt. That is the reason why I did not see the manuscript centrally anchored in the core focus of NeurIPS.
> > > >
> > > > The answer to Q2 again points to my above argument: Why should I care about convergence of the outer minimization when we aim for good generalization on future data? Generalization is particularly critical for high dimensions as sparsity in high dimensional statistics has shown in the last three decades.
> > >
> > > We acknowledge the reviewer's concern regarding the need for generalization in a class of ML problems and appreciate the need for _controlled approximations_. In fact, we suggest _adding_ the word "principled" to qualify _controlled approximations_.
> > >
> > > In our problem, generalization on future data follows naturally in a principled manner in the following way:
> > >
> > > For a set $\mathcal{M}$ and $n$ balls, our algorithm yields an optimal covering radius $\epsilon(\mathcal{M}, n)$ and centers $(x_1, \ldots, x_n)$. If future data slightly alters the set to $\mathcal{M}'$, evaluating $\mathcal{G}(x_1, \ldots, x_n)$ on $\mathcal{M}'$ or running significantly fewer iterations of the outer solver with the existing centers as a "warm start" provides a *near-optimal* cover for the new set. Hence, this is a natural and principled way to ensure generalization under small perturbations.
> > >
> > > As an *illustrative* experiment, we **perturb** the minor axis of the 2D ellipse example from **2 to 2.1**,denoting the new set as $\mathcal{M}'$. For the original $\mathcal{M}$, we have $\epsilon(\mathcal{M}, 4) = 1.6676$ with centers $(x_1, x_2, x_3, x_4)$. We compare evaluating the inner problem $\mathcal{G}(x_1, x_2, x_3, x_4)$ on $\mathcal{M}'$ versus running Algorithm 1 anew on $\mathcal{M}'$ for 150 iterations.
> > >
> > > | | Proposed generalization method | Algorithm 1 performed anew for $\mathcal{M}'$   |
> > > |:---------------------------:|:-------------------------------------------:|:----------------------------------------------------:|
> > > | Covering radius obtained $\epsilon(\mathcal{M}',4)$ |                    1.7304                   |                        1.7298                        |
> > > | Number of calls to $\mathcal{G}(x_1,x_2,x_3,x_4)$ |     1     |     4800    |
> > >
> > > We observe that evaluating $\mathcal{G}(x_1,x_2,x_3,x_4)$ once on the perturbed set with the already available centers $(x_1,x_2,x_3,x_4)$ is significantly faster than running Algorithm 1 afresh. Yet, it provides a near-optimal estimate of the new covering radius. If accepted, we will clarify regarding the generalizability for new data and include such an example in the revision.
> > >
> > > Please note that good outer convergence is critical for generalizability using our proposed scheme. To demonstrate our quality and efficiency, we have compared our numerical gradient-based outer minimization that exploits the Lipschitz continuity of $\mathcal{G}$ proven in Theorem 3.2 against other techniques like simulated annealing that do not leverage this regularity and tabulated the results in our rebuttals to Reviewer 2 (Reviewer 3wcR) and Reviewer 4 (Reviewer xqmn). We can observe from the table provided therein that schemes like simulated annealing are suboptimal even in low dimensions.
> > >
> > > We also point out that suboptimally solving the outer optimization is _not_ a principled approach to induce generalization.
> > >
> > > >The authors' answer indicates that the research is promising but somewhat premature for a presentation at NeurIPS '25.
> > >
> > > Our framework for the optimal covering problem is, to our knowledge, the first with strong theoretical guarantees under minimal assumptions (Theorems 3.2, 3.6). The theory is naturally applicable to high dimensions, and our experiments serve as illustrations of our framework. Since conferences, in general, are designed to provide platforms for vigorous exchange of new ideas so that further polished work may be submitted to journals, we firmly believe that our contributions are well-suited to this venue and respectfully disagree with your statement.
> > >
> > > We hope the preceding discussion addresses your concerns, and we request you to reconsider your evaluation. We will be available for any further clarification as needed.

---

### Decision · Program_Chairs · 2025-09-17

**Decision:**

Accept (poster)

**Comment:**

The authors provide an approximation algorithm to optimally cover a given compact subset of $R^N$ with $n$ (fixed in advance) metric balls. This is a well-known and somewhat fundamental problem. The theoretical results they provide are rigorous and use novel insights. They also include compelling experimental results, albeit somewhat limited. There is some question whether this work belongs at Neurips but ultimately it seems it would be a valuable contribution.